# Comparing Analytical Methods for Erucic Acid Determination in Rapeseed Protein Products

**DOI:** 10.3390/foods11060815

**Published:** 2022-03-12

**Authors:** Kelly Peeters, Angelica Tamayo Tenorio

**Affiliations:** 1InnoRenew CoE, Livade 6a, 6310 Izola, Slovenia; 2Andrej Marušič Institute, University of Primorska, Muzejski trg 2, 6000 Koper, Slovenia; 3Danish Technological Institute, Gregersensvej 1, 2630 Taastrup, Denmark; ante@teknologisk.dk

**Keywords:** rapeseed, protein products, erucic acid, gas chromatography mass spectrometry

## Abstract

Rapeseed meal and pressed cake are protein-rich by-products from rapeseed after oil extraction. Because of the high protein content, these by-products are an important source of food protein. Their use is motivated by the current pressure on protein prices, increasing demand for functional ingredients, and remaining controversy over wider use of soy. During process development for protein extraction from rapeseed cake or meal, special attention needs to be given to compounds such as erucic acid, which can cause problems if consumed in high amounts. Erucic acid determination is critical to ensure safety, since protein extraction procedures could lead to concentration of this compound in the final product. This research compared differences in extraction (Soxhlet and Folch) and derivatization techniques to obtain the highest erucic acid yield from rapeseed protein products. Results showed that no erucic acid accumulation occurred in the protein during its extraction from the rapeseed cake. The Soxhlet procedure was superior to Folch, as it yielded the highest concentrations of erucic acid. Furthermore, with the Folch procedure, some natural cis-configuration of erucic acid converted to its corresponding trans-configuration (brassidic acid). The latter is important, as ignoring this phenomenon can lead to underestimation of erucic acid content in rapeseed protein samples.

## 1. Introduction

Rapeseed meal and pressed cake are protein rich by-products from rapeseed after oil extraction. The average composition of rapeseed meal on dry basis consists of 30–40% crude protein, 12% crude fiber, 5–15% lipids, 4–7% ash, less than 1% calcium, and 1.2–2% total phosphorus [1]. Rapeseed cake has according to certain sources a similar protein and fat composition as rapeseed meal [2], while other sources found that rapeseed cake contains slightly lower concentration of protein while having a marginally higher fat percentage [3,4]. Because of the high protein content, these by-products are important feed sources for livestock in the EU and constitute an important source for food protein. The application in food requires processing into concentrates and isolates to ensure easy incorporation into existing product formulations. Moreover, rapeseed cake processing is motivated by the current pressure on protein prices, increasing demand for functional ingredients, and remaining controversy over wider use of soy [5,6]. During process development for protein extraction from rapeseed cake, special attention is given to the raw materials, since certain varieties of rapeseed can contain high levels of antinutritional factors such as glucosinolates, erucic acid, tannins, sinapine, and phytic acid, which can cause problems after consumption [7]. Even though rapeseed strains grown in the EU (‘00-rapeseed’) have been developed to have low levels of these antinutritional factors, their determination in final concentrated products is critical to ensure safety, given that protein extraction and concentration procedures could also lead to concentration of these compounds in the final products. This research will focus on erucic acid (fatty acid 22:1 n-9 or 22:1 ω-9), among the antinutritional factors found in rapeseed cake.

The ‘00-rapeseed’ strain is recognized to contain safe levels of erucic acid (less than 2% of the total fatty acid content) [8] and most of this compound is removed by oil extraction. However, rapeseed cake can still contain certain amounts of residual oil containing the acid [9]. Several regulations have been set up to control the maximum allowed concentration of erucic acid in food, and the latest recommendation by the European Food Safety Authority (EFSA) has lowered the maximum level of erucic acid in the fat component from 5% to 2%, proposing a tolerable daily intake (TDI) of 7 mg/kg body weight per day [10]. Based on this recommendation, determination of erucic acid in rapeseed products becomes more relevant and demands accurate quantification of this compound.

Erucic acid quantification by gas chromatography mass spectrometry (GC-MS) in food can be carried out by converting the fatty acid to its corresponding fatty acid methyl ester (FAME). This analytical tool involves several steps: (1) erucic acid extraction from the food matrix using a solvent; (2) saponification to produce salts of the free fatty acid; (3) derivatization of the free acids to form methyl esters; and (4) gas chromatography mass spectrometry (GC-MS) analysis. There is no optimal or standard protocol for steps (1) to (3). The most well-known liquid-based fatty acid extraction methods are those proposed by Folch et al. [11], using a mixture of chloroform and methanol at a ratio of 2:1 (*v*/*v*) as the extraction solvent and a final volume of 20 times the volume of the tissue sample. Modified approaches have been established over the years to improve extraction speed and reduce solvent consumption or toxicity [12]. Moreover, EFSA has compiled a list of relevant methods to extract and measure erucic acid in food products, highlighting the importance of solvent choice, such as chloroform-methanol (2:1), which can ensure breakage of complex interactions between lipids and membrane compounds (i.e., polysaccharides and proteins) [10]. Other methods for lipid extraction from a food matrix include Soxhlet extraction, which is one of the most commonly used techniques because of its straightforward use, and less conventional methods include microwave-assisted extraction, supercritical fluid extraction and ultrasonic-assisted extraction [13]. Although the less conventional methods have advantages such as being fast, robust and consuming less solvents, results have indicated no significant differences in the detected fatty acid content [13]. Regarding step (3), the derivatization of fatty acid is necessary for the subsequent analysis by GC-MS and different derivatizations are achieved depending on the agent used [12]: methanolic hydrochloric acid (HCl) can solubilize certain lipids; acetyl chloride (CH_3_COCl) causes sample spill due to exothermic reactions; sulfuric acid (H_2_SO_4_) is oxidative and unsuitable for polyunsaturated fatty acids; boron trifluoride (BF_3_) provides efficient derivatization, but forms artefacts due to its instability; basic derivatization is not suitable for free fatty acids, despite its many advantages; and trimethylsulfonium hydroxide (TMSH) has a low derivatization efficiency for polyunsaturated fatty acids.

Due to these analytical challenges, different protocols are reported for erucic acid determination in rapeseed and canola samples, and many of these protocols relate to standard procedures for extraction (ISO5509:2000 [14]; AOCS [15]; ISO 659:1998 [16]) and derivatization (ISO 5509:2000 [14]; IUPAC, 1987 [17]) of fatty acids to obtain FAMEs. Examples of the reported protocols include: hexane extraction of rapeseed seeds, derivatization with methanol acidified with 1% H_2_SO_4_ followed by GC-MS [18]; rapeseed oil refluxing in a methanol-HCl solution and measuring concentrations by gas–liquid chromatographic analysis (GLC) [19]; petroleum ether or heptane extraction of rapeseed oils and crushed seeds, derivatization with methanol and sodium methylate [20,21,22]; derivatization of oil samples with sulfuric acid/methanol, solubilization in hexane before gas chromatography equipped with a flame ionization detector (GC-FID) [23,24]; hydrolysis of rapeseed oil with methanolic KOH and analysis of the FAMES by GLC [25]; transmethylation of fatty acids in diethyl ether solution in the presence of methyl acetate, followed by reaction with 1 M sodium methoxide in methanol before GC analysis [26]; Soxhlet extraction of crushed seeds with petroleum ether, derivatization with sodium methoxide in n-hexane before GC-FID analysis [27]; petroleum ether extraction of crushed seeds in a Twisselman apparatus, derivatization with sodium methanolate methanol solution in petroleum ether followed by GC [28]; seeds mixed with anhydrous sodium sulphate, Soxhlet extraction with petroleum ether, derivatization with H_2_SO_4_ in anhydrous methanol at 100 °C, liquid–liquid extraction with petroleum ether followed by GC [29]; petroleum ether extraction of seeds in a Twisselman apparatus; and derivatization with sodium methylate in n-heptane, liquid–liquid extracted with water and acidified with HCl before GC [30]; Soxhlet extraction of seeds in hexane followed by derivatization according to IUPAC 1978 [17,31]; petroleum ether extraction of rapeseed seeds, derivatization with methanol acidified with HCl followed by GLC [32]. Besides solvent extraction and derivatizations protocols, other analytical methods have been reported: Raman spectroscopy and chemometric analysis, correlating the results to GC [33]; single bounce attenuated total reflectance (SB-ATR) Fourier transform infrared (FTIR) spectroscopy [34]; and Near Infrared Reflectance Spectroscopy (NIRS) [35,36].

The divergence in the above-described protocols makes selection of the most appropriate protocol for erucic acid quantification difficult. Even though the protocols have been developed for the same type of material (rapeseed and rapeseed oil), they still differ in extraction procedures, nature of the solvents, derivatization method and measuring method. Most of these protocols have been tried in only one laboratory or are rarely tested thoroughly or compared to each other. Therefore, this research has prepared a comparison between different extraction and derivatization techniques to obtain the highest erucic acid yield from rapeseed samples and rapeseed products, describing the challenges observed and guiding how to interpret inconsistent or different results when the same sample is analyzed by different methods. Special attention is given to differentiate between the methyl ester of erucic acid (cis 22:1 ω-9), cetoleic acid (cis 22:1 ω-11) and brassidic acid (trans 22:1 ω-9), because erucic acid and brassidic acid are cis/trans isomers, and cetoleic acid is a structural isomer that has the same molecular weight but different position of the double bond; thus, these molecular similarities are critical during quantification of only erucic acid with GC-MS.

## 2. Materials and Methods

### 2.1. Materials

Extraction was carried out with chloroform, hexane (Honeywell, analysis grade, ≥99%), and methanol (Honeywell, HPLC grade). The solvents used for derivatization were H_2_SO_4_ (Honeywell, Puriss, 95–97%), BF_3_ (Sigma Aldrich, 14% in methanol), HCl (Honeywell, Puriss, ≥37%), NaCl (Carlo Erba, reagent grade), sodium hydroxide (NaOH, Honeywell, reagent grade, anhydrous pellets, ≥98%), and potassium hydroxide (KOH, Honeywell, reagent grade).

Qualification of the FAMEs methyl erucate, methyl cetalaicate and methyl brassidate were performed with the marine source analytical standard, polyunsaturated fatty acid mix n^o^1 (Pufa n°1) and quantification standard for transfats (Restek-35629).

The derivatized fats were measured on an Agilent 7890B gas chromatograph (GC) coupled to an Agilent 5977B mass spectrometer (MS). The instrument was equipped with a Gerstel autosampler. A Restek Rt-2560 column (100 m × 0.25 mm ID with 0.20 µm film thickness) was used.

### 2.2. Samples

The samples analyzed in this study are protein products obtained from the BBI-JU project Pro-Enrich (grant agreement number. 792050), which has worked on pilot-scale process development of protein extraction from rapeseed cake by the Biorefinery Pilot Plant of the Danish Technological Institute (DTI), Taastrup, Denmark. The protein products were obtained from cold-pressed rapeseed cake (CPR) or hot-pressed rapeseed cake (HPR), using the following unit operations: aqueous extraction process, separation and filtration, and drying. Figure 1 presents a general process diagram. For the extraction, the rapeseed cakes at different solid-to-water ratios were subjected to various processing conditions (Table 1). A decanter centrifuge was employed for solid/liquid separation. Membrane filtration was used for liquid/liquid separation (i.e., 0.2 µm microfiltration) and concentration (i.e., 10 kDa ultrafiltration or 300 Da nanofiltration). Demineralized water was used for diafiltration when indicated. The final protein products were dried by different methods, resulting in fine powders. These powders are denoted as flour, concentrate or isolate depending on the crude protein content. All products were produced only once as part of up-scaling activities for industrial process development. Therefore, samples have no replicates.

### 2.3. Extraction and Derivatization of Rapeseed and Protein Products

Conventional types of lipid extraction and derivatization methods were used to extract erucic acid from the protein products and subsequently transform it to its methyl ester for analysis. For lipid extraction, the Folch method was compared with Soxhlet extraction. Hexane was selected for Soxhlet extraction based on Laroche et al. [37], who found that hexane has the same extraction efficiency as petroleum ether, and performs better than ethanol regarding relatively non-polar fatty acids. For derivatization, acids, bases and BF_3_ were used for comparison given the disadvantages and limitations described in the introduction.

First, extraction method 1 (i.e., Folch method) was combined with the three derivatization methods (samples were prepared in duplicates). The derivatization method that resulted in detection of the highest concentration of erucic acid, in either the natural, cis-isomer, or the transformed, trans-isomer (brassidic acid—both were successfully detected in the samples obtained by the Folch extraction, and the measured concentrations were similar among the replicates) was consequently used for the samples from the second extraction method (i.e., Soxhlet method). As all the solvents were removed after each of the extraction procedures, it is safe to assume that if both isomers were successfully derivatized, and subsequently detected with this derivatization method for one extraction procedure, this would be true for the other one, too. Samples were prepared in duplicate also for the extracts from the second extraction method.

#### 2.3.1. Lipid Extraction

Method 1 was based on Folch [11] and taken from Araujo et al. [38]: 0.2 g of sample (protein powder or rapeseed cake) was weighed and mixed with 4 mL of chloroform: methanol 2:1 (*v*/*v*) and internal standard. The mixture was shaken for 30 s and left at −20 °C overnight. Then, salt-saturated water was added. The organic phase was collected after biphasic separation. The sample was filtered and the solvent was nitrogen-evaporated to dryness.Method 2: 0.2 g of sample (protein powder or crushed rapeseed) was extracted with 50 mL n-hexane at 60 °C for 4 h. The solvent was removed via vacuum evaporation with a rotavapor.

#### 2.3.2. Derivatization

Method 1 was based on Yi et al. [39]. Extracted lipids, which are dissolved in 0.25 mL chloroform, were methylated with 1.5 mL 5% anhydrous HCl/methanol (*w*/*v*). The mixture was heated to 80 °C for 1 h. After cooling to room temperature, a few drops of water and 2 mL hexane were added. The hexane layer was collected and reduced in volume to 0.2 mL before measurement.Method 2 was based on Omidi et al. [40]. Extracted lipids were saponified with 5ml of methanolic NaOH (0.5 M) solution by refluxing for 15 min at 100 °C. After addition of 2.2 mL BF3 (12–15%) in methanol solution, the sample was heated for 5 min at 100 °C. The FAMEs were extracted from a 2 mL NaCl saturated mixture in water, with 2 × 1 mL hexane (centrifuge 3000 rpm). The hexane fraction was concentrated to a final volume of 0.2 mL.Method 3 was based on Brotas et al. [41]. First, 3 mL of a 0.6 M KOH in methanol was added, followed by stirring for 10 s. The tube was purged with a gentle nitrogen flow, to remove air and prevent oxidation of the compounds. The solution was heated to 70 °C in a water bath for 10 min and shaken twice during this heating. After the oil droplets had disappeared, 3 mL of a 5% solution of H_2_SO_4_ in methanol was added and the mixture was cooled; the tube was purged again with nitrogen and then heated to 70 °C in a water bath for 5 min. After this, 2 mL of a saturated solution of NaCl and 2 mL of hexane were added, the tube was shaken, and the mixture was centrifuged at 4000 rpm for 10 min. The organic phase was collected and concentrated to 0.2 mL volume.

### 2.4. GC-MS Measurements

For the GC-MS measurements, 1 µL of each derivatized sample was injected (injection temperature 200 °C, splitless injection). The column flow was set to 1.2 mL/min using helium as the carrier gas. The temperature program started with a temperature of 120 °C held for 2 min, a ramp of 5 °C per minute to 160 °C, a ramp of 2 °C per minute to 210 °C, followed by a ramp of 1 °C to 240 °C and held for 2 min. The transfer line, ion source, and quadrupole analyzer temperatures were maintained at 290 °C, 230 °C, and 150 °C, respectively. A solvent delay of 10.0 min was selected. In the full-scan mode, electron impact ionization (EI) mass spectra in the range of 40–550 *m/z* were recorded at 70 eV electron energy.

### 2.5. Data Analysis

Data analysis and integration of peak areas were carried out with Agilent Masshunter Workstation software. Qualification of methyl esters of erucic acid, brassidic acid and cetoleic acid was performed with FAME standards based on their retention time and fragmentation pattern. Quantification of all three FAMEs of interest was based on appropriately diluting the methyl brassidate standard, which had a certified concentration. A calibration curve was prepared by diluting the standard to concentrations of 0.0, 0.6, 3, 6, 30, 60 and 300 µg/mL. Transformation from FAMEs concentrations to the respective fatty acid concentrations in rapeseed and rapeseed protein extracts was performed by using a conversion factor based on their corresponding molecular weights. Concentrations were recalculated to express µg of fatty acid per g of protein product. The results of 2 replicates of each sample were averaged and the standard deviation was given. As no significant differences among the replicates were observed, the procedure was evaluated as successful and accurate, and investigating further replicates was not necessary.

## 3. Results

The experimental sequence was performed as follows: First, it was checked if it was possible to differentiate retention time during GC-MS measurements between the standards for all three isomers of the erucic acid methyl ester (methyl erucate, brassidate and cetoleidate). Then, these compounds were determined in the various protein products, and extracted by the Folch procedure combined with different derivatization. It was somewhat surprising to find that the naturally occurring cis-isomer of the erucic acid was also converted partially to its trans-isomer, regardless of the derivatization process used. The best derivatization method (this is the highest concentration of the sum of both detected isomers methyl erucate and brassidate) was used to test the same protein products, but extracted with the Soxhlet process. This approach turned out to be optimal, as the highest concentration of erucic acid was obtained, and no conversion to the trans-isomer was detected.

As mentioned above, the GC-MS procedure was optimized with the use of standards to separate the cis/trans isomers methyl erucate (cis 22:1 ω-9) and methyl brassidate (trans 22:1 ω-9), and their structural isomer methyl cetalaicate (cis 22:1 ω-11). Because of their molecular similarities, these FAMEs cannot be separated on a standard column. Therefore, individual cis and trans isomers were resolved on a 100 m column with a highly polar biscyanopropyl phase, which gives the selectivity needed for resolving FAME isomers [42]. The results can be found in Figure 2.

After extraction and derivatization, the FAME composition was determined in rapeseed cake and rapeseed protein products. The retention times from each compound in the standard were used to identify the peaks from the rapeseed protein products, since methyl brassidate, methyl cetoleidate and methyl erucate cannot be differentiated by their fragmentation pattern alone. The concentrations of erucic acid, cetoleic acid and brassidic acid were determined as described in Section 2.5 with a calibration curve. A high degree of linearity was obtained with R2: 0.9998. The limit of detection was 0.08 µg/g rapeseed or rapeseed protein product. Cetoleic acid was not detected in any of the samples. The concentration of erucic acid and brassidic acid, for each different combination of extraction and derivatization technique, can be found in Table 2 and Table 3, respectively.

To select the best derivatization method to be tested with extraction method 2, erucic acid and brassidic acid concentrations were added up, showing that derivatization method 1 yielded the highest erucic acid concentrations in rapeseed and rapeseed protein concentrates, while derivatization method 2 degraded a large portion of the fatty acids (data not shown, but total fatty acid concentrations were a factor of 10 lower than in the other methods) and was the least favorable of all methods. Therefore, derivatization method 1 was chosen to be the only one used with extraction method 2.

Rapeseed is a natural material, and its fatty acids are therefore expected to have a cis-configuration. Consequently, only erucic acid (cis 22:1 ω-9) is expected to be found in the analyzed samples. However, both the cis form (erucic acid) and the trans form (brassidic acid) were detected (Table 2 and Table 3), when extraction method 1 with chloroform: methanol was used. The highest cis–trans conversions were obtained when extraction method 1 was combined with derivatization method 3. On the other hand, when extraction method 2 with hexane was used, the trans form was not observed (Table 2), suggesting no cis–trans conversion. Based on these observations, the Folch method (method 1) is believed to set the conditions for transformation of a certain quantity of the cis-configuration to the trans-configuration during the subsequent derivatization process, resulting in lower erucic acid quantification (and the detection of brassidic acid).

Since extraction method 2, combined with derivatization method 1, gives in general the highest yields of the erucic acid and does not lead to cis–trans conformation switches, the percentage of erucic acid in relation to the total fatty acid content was calculated for that method. The results can be found in Table 4.

Table 4 shows that after CPR2 (lipid enriched phase) is removed, a certain amount of lipids still stays inside the final protein products. This total fatty acid content is very similar for all the protein products, as well for hot-pressed rapeseed as cold-pressed rapeseed, which means that the last processing steps (filtration and drying) have only a small effect on the total fatty acid content of the protein products. The amount of erucic acid in the final protein products, however, is significantly dependent on these filtration and drying processes. It can be seen for cold-pressed rapeseed that in sample CPR 3, erucic acid accumulated in the final product. The use of subsequent diafiltration reduces the erucic acid content substantially (CPR 4).

## 4. Discussion

Based on the EFSA recommendation of an allowed daily consumption of erucic acid of 7 mg/kg body weight, the erucic acid concentrations found in the samples are considered low; even the rapeseed cake itself is not harmful. An adult male would have to ingest at least a kilogram of any of the discussed protein products to ingest such a dose, and this is not practically possible. Products CPR 1–4 are the result of process development and optimization to ensure a high protein product from rapeseed cake, ending with protein isolate CPR 4 (91.2% protein content) with the lowest erucic acid percentage of all CPR products. CPR 3, which was a protein concentrate (57.3% protein content), had the highest erucic acid content. Different filtration technologies (e.g., diafiltration for CPR 4) proved to be far superior, increasing protein concentration and reducing erucic acid levels considerably. It can be speculated that the fatty acids are complexed to other macromolecules (e.g., carbohydrates) [43], which are removed during the diafiltration steps. Regarding CPR products, the obtained differences in erucic acid content confirm the need for erucic acid determination during process development.

Regarding products from hot-pressed rapeseed cake (HPR), the processing conditions included enzymatic treatment at different pH, temperatures and reaction times (Table 1), which can account for the difference in erucic acid content observed (Table 4). Although all concentrations are extremely low, a small increase in erucic acid content was observed in HPR 3, which had the lowest pH conditions (pH 5 followed by pH 7) and shortest reaction time (4 h). Further processing and concentration steps of HPR 3 towards an isolate type of product must use the right settings to avoid possible accumulation of erucic acid in the final product.

The observation that during derivatization following the extraction, erucic acid switches from its cis to trans form (brassidic acid) is an interesting phenomenon, knowing that only the cis form is expected in natural material such as rapeseed. This change in cis–trans configuration has been described in the literature for fatty acids containing double bonds, such as lineolic acid. This fatty acid has two conjugated double bonds and the change in cis–trans configuration is caused by prolonged exposure to strong acidic or alkali conditions [44,45,46]. In our samples, strong alkali or acidic conditions cannot be the cause of isomerization since the occurrence of this observed phenomenon is dependent on the extraction method, which does not contain any strong acids or alkalis. Chatgilialoglu et al. [47] have discussed another pathway for cis-trans isomerization via thiyl radicals. Thiyl moieties can derive from sulphur-containing compounds such as proteins or peptides and the isomerization reaction can already occur at 90−120 °C without adding an initiator [48,49]. These conditions are easily met in the sample preparation for the determination of fatty acids. Since the analyzed samples are rich in proteins, the presence of a small amount of proteins and peptides is likely being co-extracted with the fatty acids in the chloroform: methanol solution [48]. During the derivatization step, where higher temperatures are used, these extracted proteins and peptides can cause the initiation of the radical, leading to the isomerization of the fatty acids. Conversely, proteins or peptides are unlikely to be found in hexane in the Soxhlet extraction. Therefore, it is logical that the cis–trans conversion was not observed in our experiments during the derivatization when this type of extraction was used.

The proper detection of erucic acid in all the steps of feedstock transformation processes in the food of food supplement industry is paramount, as erucic acid is known to have antinutritional properties potentially harmful to human health. Extraction procedures differ in their efficiency to extract the target molecule(s), and the quality and quantity of co-extracted material. This is evident from the results of this work, as well as that the target molecules can undergo transformations during extraction and derivatization. For erucic acid, this is especially the case for its cis- to trans-isomer conversion into brassidic acid during the sample preparation phase. Therefore, special effort needs to be made to monitor all possible erucic acid isomers to ascertain the real concentration of its naturally occurring form in the original sample.

## 5. Conclusions

The Folch extraction method with chloroform can result in an underestimation of erucic acid content in rapeseed and rapeseed protein samples due to isomerization of fatty acids during sample preparation. Consequently, Soxhlet extraction with hexane is proposed as the most appropriate fatty acid extraction method for erucic acid quantification.

The use of rapeseed strains with low content of antinutritional factors, such as erucic acid, and proper processing steps with adequate parameters seem to ensure low content of this compound in refined protein products. Nevertheless, methodologically correct erucic acid determination must be considered as a critical parameter during process development since process conditions can result in fractions with erucic acid content above the recommended value.

## Figures and Tables

**Figure 1 foods-11-00815-f001:**
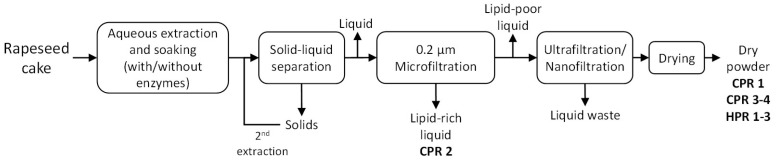
General process flow diagram for protein extraction from rapeseed cake.

**Figure 2 foods-11-00815-f002:**
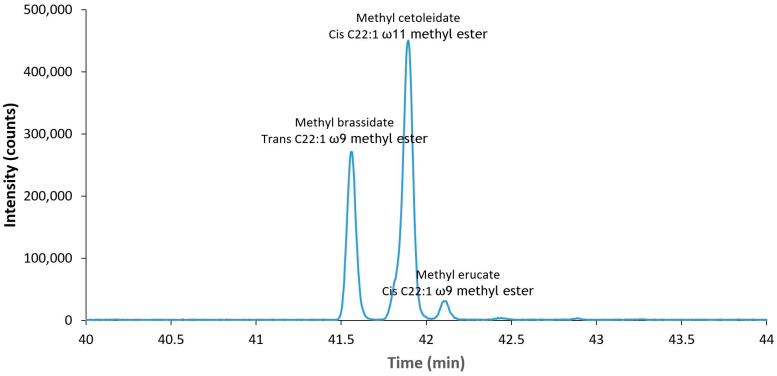
Separation of methyl brassidate, methyl cetoleidate and methyl erucate on a Restek Rt-2560, 100 m column.

**Table 1 foods-11-00815-t001:** Description of the process conditions used to obtain protein products from cold-pressed rapeseed cake (CPR) and hot-pressed rapeseed cake (HPR).

Sample Name	Process Conditions	Sample Description	Protein Content
Rapeseed cake	Rapeseed cake	Solid raw material	32.8%
CPR 1	Soaking overnight; microfiltration (filter pore size: 0.2 µm) for liquid–liquid separation and ultrafiltration (filter pore size: 5 kDa) for concentration; spray drying	Concentrate	70.8%
CPR 2	Soaking overnight; microfiltration (filter pore size: 0.2 µm) for liquid–liquid separation; oven drying	Flour	35.0%
CPR 3	Soaking overnight; microfiltration (filter pore size: 0.2 µm) for liquid–liquid separation and ultrafiltration (filter pore size: 10 kDa) for concentration, oven drying	Concentrate	57.3%
CPR 4	Soaking overnight; microfiltration (filter pore size: 0.2 µm) for liquid–liquid separation with diafiltration; ultrafiltration (filter pore size: 10 kDa) for concentration with diafiltration; freeze drying	Isolate	91.2%
HPR 1	Soaking and enzymatic hydrolysis (pH 8) overnight (T: 55 °C); microfiltration (filter pore size: 0.2 µm) for liquid–liquid separation; ultrafiltration (filter pore size: 10 kDa) for concentration; spray drying	Concentrate	58.9%
HPR 2	Soaking and enzymatic hydrolysis (pH 6.7) overnight (T: 58 °C); microfiltration (filter pore size: 0.2 µm) for liquid–liquid separation; nanofiltration (filter pore size: 300 Da) for concentration; spray drying	Concentrate	55.0%
HPR 3	Soaking and enzymatic hydrolysis (pH 5.0) for 4 h (T: 50 °C); pH adjustment to pH 7.0 (reaction time: 30 min); 0.2 µ microfiltration (filter pore size: 0.2 µm) for liquid–liquid separation; nanofiltration (filter pore size: 10 kDa) for concentration; spray drying	Concentrate	54.1%

**Table 2 foods-11-00815-t002:** Quantity of erucic acid (µg/g of sample) in rapeseed cake and protein products CPR1-4 and HPR 1-3 for different extraction (E) and derivatization (D) methods (E1: Folch method, E2: Soxhlet method; D1: HCl/methanol derivatization, D2: NaOH/BF_3_ derivatization, D3: H_2_SO_4_ derivatization—described in detail in Section 2.3) LOD is 0.08 µg/g.

Sample	Erucic Acid Concentration (µg/g of Sample) for Different Extraction and Derivatization Conditions			
	E1 + D1	E1 + D2	E1 + D3	E2 + D1
Rapeseed cake	2.61 ± 0.39	3.04 ± 0.45	17.7 ± 2.7	109 ± 8
CPR 1	1.13 ± 0.17	0.70 ± 0.11	0.33 ± 0.05	5.54 ± 0.83
CPR 2	36.5 ± 2.7	<LOD	44.4 ± 3.3	65.3 ± 5.0
CPR 3	13.0 ± 1.9	0.84 ± 0.13	0.81 ± 0.12	67.5 ± 5.1
CPR 4	0.33 ± 0.05	0.44 ± 0.07	2.67 ± 0.40	0.96 ± 0.14
HPR 1	2.17 ± 0.33	0.67 ± 0.10	0.45 ± 0.07	4.31 ± 0.65
HPR 2	4.15 ± 0.62	0.12 ± 0.02	0.67 ± 0.10	1.05 ± 0.16
HPR 3	10.3 ± 1.5	0.16 ± 0.03	1.22 ± 0.18	8.57 ± 1.29

**Table 3 foods-11-00815-t003:** Quantity of brassidic acid (µg/g of sample) in rapeseed cake and protein products CPR1-4 and HPR 1–3 for different extraction (E) and derivatization (D) methods (E1: Folch method, E2: Soxhlet method; D1: HCl/methanol derivatization, D2: NaOH/BF_3_ derivatization, D3: H_2_SO_4_ derivatization—described in detail in Section 2.3) LOD is 0.08 µg/g.

Sample	Brassidic Acid Concentration (µg/g of Sample) for Different Extraction and Derivatization Conditions			
	E1 + D1	E1 + D2	E1 + D3	E2 + D1
Rapeseed cake	41.2 ± 3.1	<LOD	3.34 ± 0.50	<LOD
CPR 1	<LOD	0.42 ± 0.06	1.69 ± 0.25	<LOD
CPR 2	<LOD	0.22 ± 0.03	<LOD	<LOD
CPR 3	<LOD	1.20 ± 0.18	1.69 ± 0.25	<LOD
CPR 4	<LOD	1.08 ± 0.16	0.31 ± 0.05	<LOD
HPR 1	<LOD	0.23 ± 0.04	2.14 ± 0.32	<LOD
HPR 2	<LOD	0.22 ± 0.03	1.17 ± 0.17	<LOD
HPR 3	0.42 ± 0.06	0.22 ± 0.03	1.21 ± 0.18	<LOD

**Table 4 foods-11-00815-t004:** Percentage (%) of erucic acid in relation to the total fatty acid content for method E2 + D1.

	Rapeseed Cake	CPR 1	CPR 2	CPR 3	CPR 4	HPR 1	HPR 2	HPR 3
Counts fatty acids (10^8^)	32.0	3.21	85.0	2.28	1.95	2.52	1.64	1.94
Counts erucic acid (10^6^)	19.8	1.01	11.9	12.3	0.17	0.78	0.19	1.56
% erucic acid	0.62	0.31	0.14	5.40	0.089	0.31	0.12	0.80

## Data Availability

Not applicable.

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
