# Peer review of "Comparing Analytical Methods for Erucic Acid Determination in Rapeseed Protein Products"

_foods, 2022, doi:10.3390/foods11060815_

Round 1

Reviewer 1 Report

The authors reported a study comparing various analytical techniques that involve different extraction and derivatization methodologies to determine the erucic acid in rapeseed protein products.  There are some serious issues regarding this manuscript for the authors to address:

  • Overall the materials and method, results, and the discussion sections of the manuscript should be improved.
  • Lines 128-130 should be moved to the materials and methods section.
  • Line 133 “The Materials and Methods” reads not right in that sentence
  • The title of Table 1 is not descriptive, it should be improved. Footnotes should be added for the abbreviated sample names.
  • Line 170 Sentence should be improved, ….methods were used for which purpose?
  • How did the authors evaluate their results? They should add a data analysis/statistics section.
  • How many times did they repeat their analysis? I can only see the replication numbers in the GC-MS part, what about the extraction, derivatization parts?
  • Lines 225-227 should be mentioned in the results section.
  • Table 2 and 3’s titles should be improved; it is difficult to read and understand.

Author Response

  • Overall the materials and method, results, and the discussion sections of the manuscript should be improved.

The text was given to another co-worker to define its weaknesses and adapted significantly during the review process.

  • Lines 128-130 should be moved to the materials and methods section.

The sentence was moved to section 2.2 Samples

  • Line 133 “The Materials and Methods” reads not right in that sentence

The authors want to thank Reviewer 1 for noticing the mistake. It was a left-over of copy-pasting the text into the template.

  • The title of Table 1 is not descriptive, it should be improved. Footnotes should be added for the abbreviated sample names.

The title of Table 1 was improved to: ‘Description of the used process conditions to obtain protein extracts from cold pressed rapeseed cake (CPR) and hot pressed rapeseed cake (HPR)’

  • Line 170 Sentence should be improved,….methods were used for which purpose?

The sentence was repaired to: Conventional types of lipid extraction and derivatization methods were used to extract erucic acid from the protein extracts and subsequently transform it to its methyl ester for analysis.

  • How did the authors evaluate their results? They should add a data analysis/statistics section.

Data analysis and integration of peak areas were done with Agilent Masshunter Workstation software. Qualification of methyl esters of erucic acid, brassidic acid and cetoleic acid was performed with FAME standards based on their retention time and frag-mentation pattern. Quantification of all three FAMEs of interest was based on appropriately diluting the methyl brassidate standard, which had a certified concentration. A calibration curve was prepared by diluting the standard to concentrations of 0.0, 0.6, 3, 6, 30, 60 and 300 µg/mL Transformation from FAMEs concentrations to the respective fatty acid concentrations in rapeseed and rapeseed protein extracts was done by using a conversion factor based on their corresponding molecular weights. Concentrations were recalculated to express µg of fatty acid per g of protein product. The results of 2 replicates of each sample were averaged and the standard deviation given. As no significant differences among the replicates were observed, the procedure was evaluated as successful and accurate, and investigating further replicates was not necessary

  • How many times did they repeat their analysis? I can only see the replication numbers in the GC-MS part, what about the extraction, derivatization parts?

The authors agree with reviewer 1. Because the data analysis section was merged with the GCMS section it seemed as if the duplicates were referring to the GCMS measurement itself. Duplicates however referred to the whole sample preparation. We originally added the sentence about duplicates to the end because we saw it as the end of the whole process. Since now we see it leads to confusion, we moved the sentence to section 2.3. Next we separated the rest of the paragraph from the GCMS analysis and called it 2.5 data analysis.

  • Lines 225-227 should be mentioned in the results section.

Lines 225-227 were moved to the results section and inserted around line 250

  • Table 2 and 3’s titles should be improved; it is difficult to read and understand.

We thank the reviewer for this remark. The Table 2 and 3’s titles were improved in the manuscript as follows:

Table 2. Quantity of erucic acid (µg/g of sample) in rapeseed cake and protein products CPR1-4 and HPR 1-3 for different extraction (E) and derivatization (D) methods (E1: Folch method, E2: Soxhlet method; D1: HCl/methanol derivatization, D2: NaOH/BF3 derivatization, D3: H2SO4 derivatization – described in detail in section 2.3) LOD is 0.08 µg/g.

Table 3. Quantity of brassidic acid (µg/g of sample) in rapeseed cake and protein products CPR1-4 and HPR 1-3 for different extraction (E) and derivatization (D) methods (E1: Folch method, E2: Soxhlet method; D1: HCl/methanol derivatization, D2: NaOH/BF3 derivatization, D3: H2SO4 derivatization – described in detail in section 2.3) LOD is 0.08 µg/g.

Reviewer 2 Report

The authors present a comparing analytical method for erucid acid determination in rapeseed protein products as tool valorization of  byproducts.   The  study evaluated two lipid extraction method, and three derivatization ones before GC-MS determination of FAME isomers in the cited matrix. The method proved to be applicable with acceptable LOD of 0.08 ug/g. This research investigated extraction and derivatization methods with satisfactory results.

Authors should add the MS identification parameters .

Author Response

Authors should add the MS identification parameters

Following sentence was added to section 2.4 GC-MS measurements: The transfer line, ion source, and quadrupole analyser temperatures were maintained at 290 °C, 230 °C, and 150 °C, respectively. A solvent delay of 10.0 min was selected. In the full-scan mode, electron impact ionization (EI) mass spectra in the range of 40–550 m/z were recorded at 70 eV electron energy.

Reviewer 3 Report

The authors studied two extraction methods and three derivatization methods and from the results presented in Table 2 and Table 3 concluded that extraction method 2 was the best since extraction method 1 causes the transformation from the cis-configuration to the trans-configuration. But at the same time they write that the combination of E1+D3 causes the highest cis-trans conversions. Also they do not present the data of E2+D2 and E2+D3. My question is are the authors so sure that the conversion is due to E1 and not the derivatization method? That why they have o present the data for E2+D2 and E2+D3 and to clarify that.

Additionally the authors have to present the intraday and interday results for the best combination E+D

Author Response

The authors studied two extraction methods and three derivatization methods and from the results presented in Table 2 and Table 3 concluded that extraction method 2 was the best since extraction method 1 causes the transformation from the cis-configuration to the trans-configuration. But at the same time they write that the combination of E1+D3 causes the highest cis-trans conversions. Also they do not present the data of E2+D2 and E2+D3. My question is are the authors so sure that the conversion is due to E1 and not the derivatization method? That why they have o present the data for E2+D2 and E2+D3 and to clarify that.

The reasoning why only one derivatisation method was used with the second extraction procedure is now better explained in the manuscript (“new” lines 333 – 343). It is very reasonable to assume that if the selected derivatisation methods yielded the best detection of both discussed isomers of erucic acid in one extraction procedure, that the same would have happened with the samples from the second extraction process. Furthermore, the intensity of the cis-trans conversions is not consequential for the findings of this paper – the intention is mainly to alert that different extraction procedures can set the conditions for this conversion, and that care needs to be taken to observe both isomers to ascertain the correct levels of the “original” erucic acid concentration in the samples. The text now reads:

“First, extraction method 1 (i.e. Folch method) was combined with the three derivatization methods (samples were prepared in duplicates). The derivatization method that resulted in detection of the highest concentration of erucic acid, in either the natural, cis-isomer, or the transformed, trans-isomer (brassidic acid – both were success-fully detected in the samples obtained by the Folch extraction, and the measured concentrations did not differ among the replicates) was in the continuation used for the samples from the second extraction method (i.e. Soxhlet method). As all the solvents were removed after each of the extraction procedure, it is safe to assume that if both isomers were successfully derivatised, and subsequently detected with this derivatisation method for one extraction procedure, that that would be true for the other one, too. Samples were prepared in duplicates also for the extracts from the second extraction method.”

Additionally the authors have to present the intraday and interday results for the best combination E+D

With all the measurements standards were used to account for possible differences in the functioning of the GC-MS apparatus. Furthermore, an internal standard was also used in all of the measurements. No discernible differences were observed; therefore the authors do not see the need to discuss either intra- or inter-day variation in measurement results.

Round 2

Reviewer 1 Report

Authors made the appropriate changes that I pointed out.

Reviewer 3 Report

The manuscript is very improved. I suggest being published in its present form.